# Modulation of Neuroendocrine and Immunological Biomarkers Following Rehabilitation in Sarcopenic Patients

**DOI:** 10.3390/cells11162477

**Published:** 2022-08-10

**Authors:** Federica Piancone, Francesca La Rosa, Ivana Marventano, Ambra Hernis, Rossella Miglioli, Fabio Trecate, Marina Saresella, Mario Clerici

**Affiliations:** 1IRCCS Fondazione Don Carlo Gnocchi, 20147 Milan, Italy; 2Department of Pathophysiology and Transplantation, University of Milan, 20122 Milan, Italy

**Keywords:** sarcopenia, rehabilitation, inflammation, cytokines, neurotransmitters, norepinephrine

## Abstract

This study aimed to investigate if rehabilitation could down-regulated sarcopenia-associated inflammation by modulating the crosstalk between the neuroendocrine and immune systems, with the aim of ameliorating quality of life of sarcopenic subjects. A total of 60 sarcopenic patients (49 females and 11 males; median age 74.5, interquartile range 71–79), undergoing a personalized rehabilitation program, have been recruited and subjected to: (1) functional and physical evaluation (Short Physical Performance Battery (SPPB), Barthel Index and Tinetti Test); (2) pro-inflammatory IL-1β, TNF-α, IL-6, IL-18, and anti-inflammatory IL-10 cytokines plasmatic level measures; and (3) norepinephrine, epinephrine, dopamine, and serotonin neurotransmitter level evaluation at time of enrollment (T0) and once rehabilitation was concluded (1 month, T1). Rehabilitation combined a balance and strength training program with two daily sessions that were fine-tuned and personalized according to the ability of the patient. The results showed a significant increase at T1 in the plasmatic levels of IL-10 (*p* = 0.018) and of norepinephrine (*p* = 0.016)), whereas the concentration of IL-18 was significantly reduced (*p* = 0.012). Notably, changes in norepinephrine were positively correlated with clinical improvements (Tinetti and Barthel scores, *p* ≤ 0.0001; SPPB scores, *p* = 0.0002). These results show that efficient rehabilitation induces a reduction of inflammation, suggesting that this effect could be mediated by a modulation of the neuro-immune axis that results in an increase of norepinephrine.

## 1. Introduction

Sarcopenia is a geriatric generalized skeletal muscle disorder that involves the accelerated loss of muscle mass and strength/function during aging and represents a major risk factor for adverse health-related events in later life [1], and a significant social and economic burden [2]. Based on these premises, it is important to clarify the working mechanisms underlying this disease.

The prevalence of sarcopenia is highly variable (3–24%), due to the heterogeneity of the criteria that are used to formulate its diagnosis [3]. There are numerous causes of sarcopenia and can include inactivity, chronic diseases, inflammation, insulin resistance, nutritional deficiencies, and cytokine and neuroendocrine imbalance, among others [4,5]. Chronic low-grade inflammatory profile has been recently suggested to contribute to sarcopenia, by affecting both muscle protein breakdown and the central nervous system (CNS). Over the past decades evidences have, in fact, clearly demonstrated a crosstalk between the nervous and the immune systems that communicate with each other through shared chemical messenger molecules including hormones, neurotransmitters, or cytokines [6].

In the initial phase of inflammatory processes, the CNS adopts an “inflammatory configuration” with the activation of the sympathetic nervous systems (SNS), characterized by the increase of the hypothalamic-pituitary-adrenal (HPA) axis activity [7]. This activation leads to the release of sympathetic neurotransmitters, including norepinephrine (NE), that have direct influence of immune cells. In this early phase, this process is strictly controlled: the sympathetic nervous system, while inducing the differentiation of adaptive immunity cells toward a pro-inflammatory phenotype, by the production of pro-inflammatory TNF-α, IL-6, and IL-1β cytokines [7], also inhibits innate immune cells by the stimulation of β2- adrenergic receptors (β2-ARs). Although, the net outcome of SNS influence in this phase is pro-inflammatory. If inflammation becomes chronic, the overall response changes into a “chronic inflammatory condition” that is characterized by increased activity of the HPA axis without immunosuppression (glucocorticoid receptor desensitization and inadequacy).

Chronic low-grade inflammation is observed in age-related reorganization of the neuromuscular system and is increased by sedentarism [8,9,10]. Systemic low-grade inflammation, defined as a two- to four-fold elevation in the circulating levels of pro-inflammatory cytokines, is considered as an underlying mechanism of aging and age-related diseases. Several studies have shown that, in the elderly, high levels of inflammatory cytokines were correlated with a reduction in muscle strength and loss in muscle mass. Conversely, it is thought that exercise exerts anti-inflammatory effects, but the mechanism regulating the immunomodulatory effects of exercise on inflammatory cytokine production remains to be clarified.

Moderate-to-vigorous physical exercise stimulates the release of catecholamines (CATs) epinephrine, norepinephrine, and dopamine; these mediators play an important regulatory and modulatory action by affecting metabolic processes and the immune system. Epinephrine and norepinephrine are the fast-acting “fight or flight” hormones that are produced by the adrenal medulla; their release begins upon stressful situations, but they are also released during exercise. Cortisol and epinephrine are increased during exercise as a consequence of the activation of the hypothalamic–pituitary–adrenal (HPA) axis and the sympathetic nervous system (SNS); cortisol is known to be a potent anti-inflammatory hormone [11], while catecholamines are known to be able to downregulate TNF-α- and IL-1β-driven inflammatory responses that are induced by LPS [12]. Currently, no pharmacological agents for sarcopenia are available, and the main treatment of the disease is physical therapy for muscle strengthening and gait training. To investigate whether physical therapies for sarcopenia modulate the inflammation that is associated with sarcopenia, and to investigate which mechanisms drive this effect, we analyzed pro (IL-1β, TNF-α, IL-6, IL-18) and anti (IL-10)-inflammatory cytokines, as well as the neurotransmitters norepinephrine, epinephrine, dopamine, and serotonin, in sixty sarcopenic patients who underwent a specifically-designed rehabilitation program, at the beginning and at the end of the rehabilitation.

## 2. Materials and Methods

### 2.1. Patients Enrolled in the Study

This study included sixty severe sarcopenic patients (EWGSOP diagnostic criteria) [13] whose clinical and demographic characteristics are presented in Table 1.

MMSE mini mental state examination.

The data for age and SPPB are reported as the median and interquartile range (IQR).

The data for MMSE, right handgrip, and left handgrip is reported as media ± standard deviation.

The cohort population was hospitalized and recruited at the Palazzolo Institute, IRCCS Fondazione Don Carlo Gnocchi ONLUS, Milan. During hospitalization, sarcopenic subjects underwent a rehabilitative treatment that included: (1) in the morning, 40 min, and (2) in the afternoon, 30 min sessions with assisted mobilization, progressive muscle strength training, progressive loading, exercises to improve balance and coordination (standing work proprioceptive postural balance), and walking training firstly with an assisted way and then without. Each patient was evaluated for Short Physical Performance Battery (SPPB) [14], Barthel Index [15], and the Tinetti Balance Test [16,17] before and after rehabilitation.

The study was authorized by the Ethical Committee of IRCCS Don Gnocchi Foundation (n#9_04/04/2018).

### 2.2. Plasma Sample Collection

Peripheral blood samples were collected in EDTA-containing vacutainer tubes (Becton Dickinson and Co., Rutherford, NJ, USA), and were centrifuged at 2000× *g* for 10 min to obtain plasma and stored at −80 °C until testing.

### 2.3. ELISA

Plasma IL-1β, TNF-α, IL-6, IL-18, and IL-10 concentrations were detected by Quantikine Immunoassay (R&D Systems, Minneapolis, MN, USA). The optical densities (OD) were determined with a Sunrise plate reader (Tecan, Mannedorf, Switzerland). The measured absorbance is proportional to the concentration of cytokines (IL-1β, IL-18, IL-6, TNF-α, IL-10) that are present in the plasma. All the experiments were performed in duplicate.

Neurotransmitter levels were quantified in plasma by sandwich immunoassay following the manufacturer’s instructions. Tricat TM ELISA kit (#RE59395; IBL International; Hamburg, Germany) was used to quantify norepinephrine, epinephrine, and dopamine levels. The amount of serotonin was performed by abcam Serotonin ELISA Kit (#ab133053; abcam; Cambridge, UK). The measured absorbance is proportional to the concentration of neurotransmitters (norepinephrine, epinephrine, dopamine, and serotonin) that are present in the plasma expressed in pg/mL.

### 2.4. Statistical Analysis

The data were analyzed using the statistical software MedCalc (MedCalc Software bvba, Mariakerke, Belgium). Descriptive statistics (median ± interquartile range or mean ± standard deviation, as appropriate) were used for the characterization of the study. Correlations were analyzed by Spearman’s correlation coefficient (R*sp*). Differences were considered significant at *p*-values ≤ 0.05.

## 3. Results

### 3.1. Rehabilitation Significantly Improves Clinical Parameters in Sarcopenic Subjects

The clinical outcome measures at baseline (T0) and post-rehabilitation interventions (T1) were analyzed in all the subjects that were included in the study. The results showed a significant improvement of the scores of the Tinetti test (Figure 1A), Barthel Index (Figure 1B), and Short Physical Performance Battery (SPPB) (Figure 1C) at the end of the protocol (*p* < 0.001 in all cases).

### 3.2. Rehabilitation Has an Anti-Inflammatory Effect in Sarcopenic Subjects

The concentration of the pro-inflammatory IL-1β, TNF-α, IL-6, and IL-18 cytokines and of the anti-inflammatory IL-10 cytokine was evaluated in plasma of sarcopenic subjects before (T0) and at the end (one month later, T1) of the rehabilitation program. The results showed that although no differences were observed in IL-1β (Figure 2A), TNF-α (Figure 2B), and IL-6 (Figure 2C), IL-18 production was significantly decreased after rehabilitation at T1 (*p* = 0.012) (Figure 2D). In line with this result, the concentration of IL-10, a potent anti-inflammatory protein, was significantly augmented in the plasma of sarcopenic subjects after rehabilitation (*p* = 0.018) (Figure 2E).

### 3.3. Rehabilitation Results in Neuromodulation in Sarcopenic Subjects

The concentration of neuroendocrine factors epinephrine, norepinephrine, and serotonin were analyzed in the plasma of sarcopenic patients before (T0) and after the rehabilitation program (T1). The results showed that, although no differences were observed in epinephrine (Figure 3A), dopamine (Figure 3B), and serotonin (Figure 3C), a significant increase was observed in norepinephrine after rehabilitation at T1 (*p* = 0.0016) (Figure 3D).

### 3.4. Correlation of Clinical Functional Scales and Biological Parameters

Possible correlations between the scores of the Tinetti test, Barthel Index, and the Short Physical Performance Battery (SPPB) (total scores and walking speed, chair stand, and standing balance scores) and the plasma concentration of anti-inflammatory and pro-inflammatory cytokines and of neuroendocrine factors were analyzed next. The results showed the presence of a significant positive correlation between norepinephrine concentration and both the Barthel Index (*p* < 0.0001, R*sp* = 0.51) and the Tinetti test scores (*p* < 0.0001, R*sp* = 0.53). Notably, changes in norepinephrine concentration were also significantly positively correlated with the SPPB test scores (SPPB total scores: *p* = 0.0002, R*sp* = 0.38; SPPB walking scores: *p* = 0.01, R*sp* = 0.3; SPPB sit to stand scores: *p* = 0.0001; R*sp* = 0.4; SPPB balance score *p*= 0.007; R*sp* = 0.34) and negatively correlated with IL-18 concentration (*p* = 0.015, R*sp* = −0.3). No other correlations were found (Table 2).

## 4. Discussion

Sarcopenia was initially used to describe an age-related loss of muscle mass and function [18], alone, without reference to function, but today muscle function is included in the concept of sarcopenia [1]. With an increase in the number and proportion of elderly in the population, sarcopenia is a growing global health concern due to its impact on morbidity, mortality, and healthcare expenditure [19].

Many factors, such as hormonal changes, sedentarism, malnutrition, and neuronal changes, have been suggested to contribute to the disease [20], but lately, the possible role of inflammation has gained much attention in the pathogenesis of this disease. Thus, the increased production of IL-6 and TNF-α have been associated with the increased risk of muscle mass and strength loss [21].

The dysregulation of inflammatory cytokines may be favorably altered by lifestyle choices. Epidemiological and clinical data demonstrate the positive influence of regular physical activity [22]; in a study that was conducted in a large cohort of subjects over 65 years old it has been shown that higher levels of physical activity were associated with lower serum concentrations of several markers of inflammation, suggesting that a higher degree of physical activity is associated with reduced inflammation [22].

Given these premises, we investigated whether rehabilitation: (1) could have a beneficial impact on sarcopenic subjects, and (2) could decrease sarcopenic-associated inflammation.

The results showed that rehabilitation resulted in a significant amelioration of physical conditions as suggested by the improvement in Barthel Index, Tinetti score, and Short Physical Performance Battery scores. This result suggests that regular practice of exercise, that is promoted for its positive impact in several chronic inflammatory diseases, especially cardiovascular diseases [23,24], could be beneficial in sarcopenic subjects. Moreover, clinical amelioration was accompanied by augmented IL-10 together with diminished IL-18 levels. These results are in line with previous observations suggesting that exercise is associated with a reduction in inflammatory cytokines production [25], including IL-18 [26].

Cross-sectional studies have suggested that physical exercise protects against diseases that are associated with chronic low-grade system inflammation [27], as such exercise stimulates the increase of IL-10 and inhibits the production of pro-inflammatory cytokines IL-1β [28] and TNF-α [29,30].

Furthermore, by flow-cytometry analysis it has been shown that a short single bout of 20 min moderate treadmill exercise is effective in reducing the concentration of TNF-α by monocytes [31]. This effect is possibly mediated by increased catecholamine levels via β2-adrenergic receptors seeing as how the treatment of cells with β2-adrenergic antagonists exercise effect disappears when the cells are treated with β2-adrenergic antagonists. Indeed, in general, adrenergic signaling has an immunosuppressive property in nature and has been reviewed extensively [32].

Nevertheless, the mechanism regulating the immunomodulatory effects of exercise on inflammatory cytokines remains to be clarified, thus, we investigated if this effect would be mediated by the crosstalk between immunological and neuroendocrine parameters in sarcopenic subjects.

Only 25 years ago, a functional interaction between the central nervous system (CNS) and the immune system was first proposed. Today, several bidirectional communication pathways have been described between these two systems. Nowadays it is well accepted that activated immune cells patrol the normal CNS and that products of these cells exert both protective and detrimental influences on CNS homeostasis.

Among these cellular products, norepinephrine (NE), the chemical messenger of the sympathetic nervous system, it is known to stimulate immune cell readiness during infection and immune challenge. In particular, recent analyses that were performed in animal models have demonstrated that NE downregulates the production of the pro-inflammatory mediators TNF-α, IP-10 (IFN-γ induced protein 10), and IL-1β as well as that of reactive oxygen species (ROS), while increasing the generation of the anti-inflammatory IL-10 cytokine [33]. NE binding to its adrenergic receptors (ARs), in fact, induces cyclic AMP and protein Kinase A (PKA) activation, which reduces pro-inflammatory cytokine production by the inhibition of NF-κB nuclear translocation [34].

These results were confirmed by human studies in which healthy volunteers were randomized to a 5-h intravenous infusion of either low dose NE (0.05 mg/kg/min), vasopressin (0.04 IU/min), or placebo (saline) starting 1 h before intravenous administration of 2 ng/kg LPS. As expected, LPS administration resulted in a potent response that was characterize by increased levels of TNF-α, IL-6, IL-8, IP-10, MCP-1, G-CSF (granulocyte colony–stimulating factor), and IL-10 cytokines. Notably, though, pretreatment with NE significantly increased IL-10 production without modulating that of the pro-inflammatory cytokines [33].

Furthermore, rehabilitation resulted in a significant increase in the NE concentration that was negatively correlated with IL-18 production, suggesting that the anti-inflammatory effect of rehabilitation is partly neuromodulated by NE and by the crosstalk between the neuro-immune axis.

Taken together these results confirm the efficacy of rehabilitation in reducing sarcopenia-associated inflammation and suggest that this effect could be mediated by the increase of NE. Therapeutic strategies that are aimed at increasing the concentration of NE (for example the administration of inhibitors of the reuptake of norepinephrine or exercise) could help limiting inflammatory events and have a beneficial role in sarcopenia.

## Figures and Tables

**Figure 1 cells-11-02477-f001:**
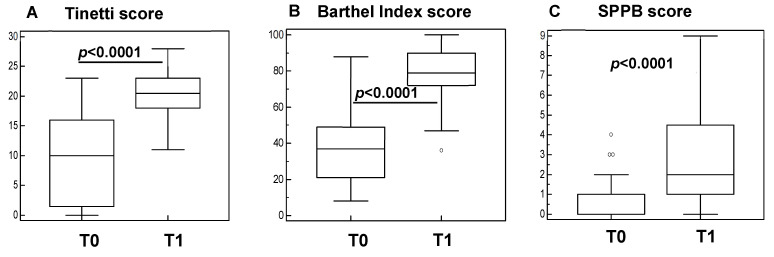
Clinical scores in sarcopenic subjects before (T0) and after rehabilitation (T1). Summary results are shown in the bar graphs. The boxes show the lower quartile, median (line across the boxes), and the upper quartile values. The whiskers represent the extreme values. The outside values are presented as separate points. Statistical significance is shown.

**Figure 2 cells-11-02477-f002:**
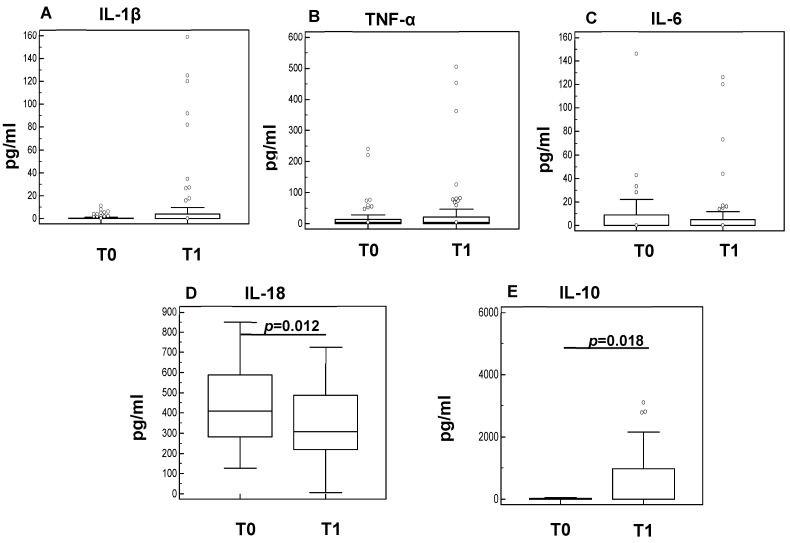
Plasmatic interleukin level at baseline (T0) and one month later at the conclusion of the rehabilitation protocol (T1). The boxes indicate the median and the first and third quartiles. Outliers are presented as dots; statistical significance (Mann–Whitney test) is indicated.

**Figure 3 cells-11-02477-f003:**
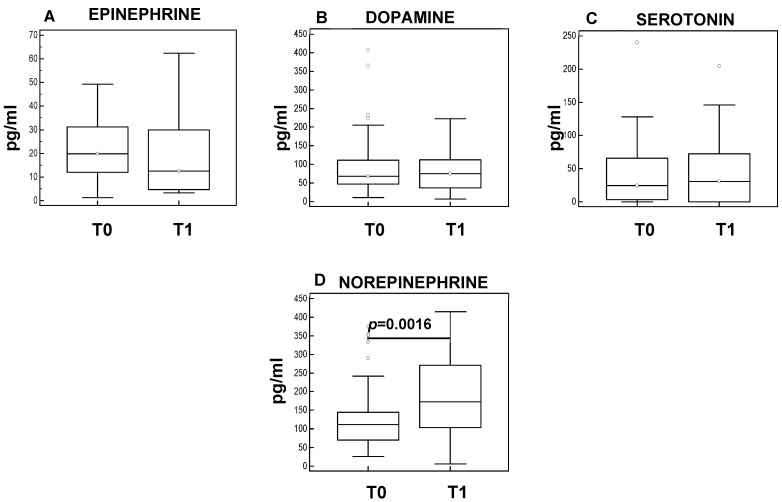
Plasma concentration before (T0) and after the rehabilitation protocol (T1). Summary results are shown in the bar graphs. The boxes stretch from the 25 to the 75 percentiles; the line across the boxes indicates the median values; the lines stretching from the boxes indicate extreme values. Outliers are plotted as individual dots; statistical significance is indicated.

**Table 1 cells-11-02477-t001:** Demographic and clinical characteristic of the study population.

	Sarcopenic Subjects (*N* = 60)
Gender (M:F)	11:49
Age (years)	74.5 (71–79)
MMSE	27.5 ± 0.3
**Handgrip** strength (right)	18.4 ± 6.6
**Handgrip** strength (left)	16.6 ± 6.6
**SPPB**	1 (0–1)

**Table 2 cells-11-02477-t002:** Correlation table. R*sp*-values and *p*-values for the correlation analysis between clinical measures (Barthel Index, SPPB tests, and Tinetti test) and laboratory measures (norepinephrine, IL-10, and IL-18). Significant correlations are indicated in bold (*p* value ≤ 0.05 and R*sp* ≥ 0.3).

Sarcopenic Patients		NE	IL-10	IL-18	Barthel Index	SPPB Walking	SPPB Balance	SPPB Sit to Stand	SPPB Total	Tinetti
NE	R*sp* *p* value		0.136 0.180	−0.3 0.015	0.51 <0.0001	0.3 0.01	0.34 0.007	0.4 0.0001	0.38 0.0002	0.53 <0.0001
IL-10	R*sp* *p* value	0.136 0.180		−0.139 0.155	0.089 0.399	0.047 0.655	0.151 0.150	0.137 0.193	0.140 0.184	0.130 0.218
IL-18	R*sp* *p* value	−0.244 0.015	−0.139 0.155		−0.237 0.022	−0.060 0.568	−0.021 0.844	−0.116 0.271	−0.071 0.499	−0.143 0.173
Barthel Index	R*sp* *p* value	0.51 <0.0001	0.089 0.399	−0.237 0.022		0.503 <0.0001	0.494 <0.0001	0.404 0.0001	0.547 <0.0001	0.807 <0.0001
SPPB walking	R*sp* *p* value	0.3 0.01	0.047 0.655	−0.060 0.568	0.503 <0.0001		0.574 <0.0001	0.474 <0.0001	0.763 <0.0001	0.572 <0.0001
SPPB balance	R*sp* *p* value	0.34 0.007	0.151 0.150	−0.021 0.844	0.494 <0.0001	0.574 <0.0001		0.675 <0.0001	0.919 <0.0001	0.416 <0.0001
SPPB sit to stand	R*sp* *p* value	0.4 0.0001	0.137 0.193	−0.116 0.271	0.404 0.0001	0.474 <0.0001	0.675 <0.0001		0.848 <0.0001	0.403 0.0001
SPPB Total	R*sp* *p* value	0.38 0.0002	0.140 0.184	−0.071 0.499	0.547 <0.0001	0.763 <0.0001	0.919 <0.0001	0.848 <0.0001		0.528 <0.0001
Tinetti	R*sp* *p* value	0.53 <0.0001	0.130 0.218	−0.143 0.173	0.807 <0.0001	0.572 <0.0001	0.416 <0.0001	0.403 0.0001	0.528 <0.0001	

Data reported Spearman’s correlation coefficient (R*sp*) and *p*-value. NE norepinephrine, SPPB Short physical performance battery.

## Data Availability

The datasets generated and/or analyzed during the current study are available from the corresponding author on reasonable request.

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
