# Peer review of "Modulation of Neuroendocrine and Immunological Biomarkers Following Rehabilitation in Sarcopenic Patients"

_cells, 2022, doi:10.3390/cells11162477_

Round 1

Reviewer 1 Report

This prospective study investigated whether rehabilitation could down-regulate sarcopenia-associated inflammation and could ameliorate the psysical condition of sarcopenic patients. A comprehensive geriatric multidimensional evaluation (Short Physical Performance Battery (SPPB), Barthel Index and Tinetti Test) as well as measure of plasmatic concentration of pro- and anti-inflammatory cytokines and of norepinephrine, epinephrine, dopamine and serotonin neurotransmitters was performed on sixty sarcopenic patients undergoing a structured rehabilitation, at T0 (time of recruitment) and at T1 (30-days; the end of the rehabilitation). Results showed that at T1 concentration of IL-10 and norepinephrine was significantly increased (p= 0.018 and p= 0.016, respectively) whereas that of IL-18 was significantly reduced (p= 0.012). Notably, changes in norepinephrine were positively correlated with improvements in the Tinetti (p< 0.0001), Barthel (p= <0.0001) and SPPB (p=0.0002) scores. Results herein show that effective rehabilitation results in a reduction of inflammation, and identify the peripheral immunological and neuroendocrine biomarkers which are modulated by rehabilitation. This study is novel and well written.

Reviewer 2 Report

The concept of the study is interesting. Sarcopenia is an important disease that deserves well investigation. The major drawback of this article is lack of a control group. We are not able to know whether the change is random or is the effect of rehabilitation. Furthermore,  the demographics demonstrated in Table 1 seemed too rough. The key measurements for sarcopenia were not listed.   

Reviewer 3 Report

The manuscript by Piancone et al. investigates the alterations in neuroendocrine and immunological biomarkers in sarcopenia patients during rehabilitation. Overall, the topic of this manuscript is interesting and needed as the contributions of these biomarkers to sarcopenia and its reversal during rehabilitation are poorly understood. The manuscript reads well. However, certain issues require attention to obtain a high-quality manuscript. Additionally, the manuscript has some syntax errors, typos, and grammatical mistakes. The following comments are aimed at improving the quality of this manuscript.

Major comments:

Abstract:

I understand that the primary aim of rehabilitation is to reduce inflammation and sarcopenia. However, it is not clear from the abstract, which molecular markers are pro- and anti-inflammatory.

Add details about the type of rehabilitation.

Ages and genders of sarcopenic patients are missing.

L-21, add comma after T1, make it concentrations and were.

L-22, add comma after bracket.

Introduction:

It generally reads well, and you nicely build the case for the study. However, syntax errors exist, commas are missing, and the sentences are on occasion, too lengthy. Please rewrite this to improve the quality.

Methods:

Please provide more details about the blood collection. Was it done immediately after the last exercise sessions or a considerable between exercise and blood collection exist? Epinephrine and nor-epinephrine have very short half-lives and the timing of blood collection is critical for correct measurements. What times of the day were those samples collected?

Results:

Fig 1C: The baseline SPPB scores are extremely low for the population of this age. In my experience and from the literature, even the sedentary older individuals with advanced age obtain significantly higher SPPB scores than your subjects (please see PMID: 33781211). Most of your subjects have SPPB scores of zero and 1 and are likely functionally dependent. I’m surprised that these patients could perform 70 minutes a day training, including resistant exercises. Please comment.

It may be interested to see if men and women respond differently to the rehab exercises. Did you find any differences in their performance and biomolecules concentrations?

Discussion:

There is almost no ‘’discussion’’ of your results here. Lines 276-325 is a brief literature review without bringing in your results in context. The remaining part of ‘’discussion’’ summarizes your results without discussing them with relevant literature. Please rewrite the whole section and discuss your results here.

Round 2

Reviewer 3 Report

The manuscript seems fine now. However, I'm still very sceptical about elderly patients with SPPB scores of 0 and 1 performing your designed exercises. 
